# Polydatin-Induced Shift of Redox Balance and Its Anti-Cancer Impact on Human Osteosarcoma Cells

**DOI:** 10.3390/cimb47010021

**Published:** 2024-12-31

**Authors:** Alessio Cimmino, Magda Gioia, Maria Elisabetta Clementi, Isabella Faraoni, Stefano Marini, Chiara Ciaccio

**Affiliations:** 1Department of Clinical Sciences and Translational Medicine, University of Rome ‘Tor Vergata’, Via Montpellier 1, 00133 Rome, Italy; alessiocimmino7@gmail.com (A.C.); magda.gioia@uniroma2.it (M.G.); stefano.marini@uniroma2.it (S.M.); 2Istituto di Scienze e Tecnologie Chimiche “Giulio Natta” SCITEC-CNR, Largo Francesco Vito 1, 00168 Rome, Italy; elisabetta.clementi@scitec.cnr.it; 3Department of Systems Medicine, University of Rome ‘Tor Vergata’, Via Montpellier 1, 00133 Rome, Italy; faraoni@med.uniroma2.it

**Keywords:** Osteosarcoma, Polydatin, ROS, GSH, ferroptosis, hypoxia, chemotherapeutic agents

## Abstract

Cancer cells demonstrate remarkable resilience by adapting to oxidative stress and undergoing metabolic reprogramming, making oxidative stress a critical target for cancer therapy. This study explores, for the first time, the redox-dependent anticancer effects of Polydatin (PD), a glucoside derivative of resveratrol, on the human Osteosarcoma (OS) cells SAOS-2 and U2OS. Using cell-based biochemical assays, we found that cytotoxic doses of PD (100–200 µM) promote ROS production, deplete glutathione (GSH), and elevate levels of both total iron and intracellular malondialdehyde (MDA), which are key markers of ferroptosis. Notably, the ROS scavenger N-acetylcysteine (NAC) and the ferroptosis inhibitor ferrostatin-1 (Fer-1) partially reverse PD’s cytotoxic effects. Interestingly, PD’s ability to hinder cell adhesion and migration appears independent of its pro-oxidant effect. Analysis of the oxidative stress regulators SIRT1 and Nrf2 at the gene and protein levels using real-time PCR and Western blot indicates an early oxidative response to PD treatment. PD remains effective under tumor-like conditions of hypoxia and serum starvation, and sensitizes OS cells to ROS-inducing chemotherapeutics like doxorubicin (DOX) and cisplatin (CIS). Importantly, PD exhibits minimal toxicity to non-tumorigenic cells (hFOB), suggesting a favorable therapeutic profile. Overall, our findings underscore that PD-induced redox imbalance plays a crucial role in its anti-OS effects, warranting further exploration into the molecular mechanisms behind its pro-oxidant activity.

## 1. Introduction

Osteosarcoma (OS) is the most prevalent form of bone tumor with a worldwide incidence of 3.4 cases per million people per year, and is one of the leading causes of cancer-related deaths during the second and third decades of life [1,2,3]. Disease treatment has evolved based on extensive clinical research. The standard approach typically involves aggressive neoadjuvant and adjuvant chemotherapy using high doses of methotrexate (MTX), doxorubicin (DOX), and cisplatin (CIS), combined with surgical removal of the tumor [4,5]. While this regimen yields a five-year survival rate of 65–70%, OS remains a major therapeutic challenge. A substantial portion of patients develop lung metastases, a leading cause of mortality, highlighting the need for alternative therapeutic strategies as current treatments fail in 20–30% of cases [2,3].

Cancer cells often adapt to oxidative stress by modulating signaling pathways that regulate the metabolism of reactive oxygen species (ROS). This adaptation supports tumor development and progression [6,7,8]. However, this same regulatory mechanism is crucial in cancer therapy, as many chemotherapeutic strategies exploit ROS manipulation to induce pro-apoptotic stimuli and target cancer cells. By increasing oxidative stress beyond the cells’ adaptive capacity, these therapies aim to trigger cell death [8,9]. Thus, controlling ROS levels is a pivotal factor both in cancer cell survival and in the effectiveness of anti-cancer treatments, as reflected in the development of drugs designed to directly or indirectly modulate ROS for therapeutic purposes [10].

Natural products have long been an important research area for the discovery of novel and bioactive molecules [11,12]. Several of these compounds have been shown to stimulate ROS production in various cancer cells, leading to various types of cell death, including apoptosis, autophagy, and ferroptosis [13]. This offers potential for anti-cancer applications with minimal toxicity to healthy tissues [14,15]. Polydatin (3,4′,5-trihydroxystilbene-3-β-d-glucoside; piceid, PD) is a stilbenoid polyphenol extracted from Polygonum cuspidatum and other dietary plants, which have a long history of use in traditional Chinese medicines [16,17]. PD is a glucoside derivative and natural precursor of resveratrol in which the glucoside group linked to position C-3 replaces the hydroxyl group of resveratrol. This structural modification gives PD a more hydrophilic character than resveratrol, leading to improved bioavailability and potentially greater therapeutic benefits [17,18,19]. Notably, PD is absorbed into cells via sodium-dependent glucose transporters (SGLT1), primarily located in the stomach and intestines [17]. The trans-isomer of PD exhibits significant therapeutic promise in a variety of medical domains, including infectious diseases, inflammatory conditions, cardiovascular disorders, and age-related pathologies like osteoporosis [17,20].

A beneficial role of PD has also been documented in the prevention and treatment of several kinds of cancers [17,21,22]. PD’s anti-cancer activity primarily involves cell cycle regulation, apoptosis induction, autophagy modulation, signaling pathway modulation, epithelial–mesenchymal transition inhibition, and regulation of oxidative stress enzymes [22,23]. Studies on the effect of PD on OS have shown a strong inhibition of cell cycle progression and cell growth. Specifically, PD promotes apoptosis by increasing the Bax/Bcl-2 ratio and reducing β-catenin signaling [24]. It also triggers autophagic cell death by lowering STAT3 expression/phosphorylation and upregulating autophagy-related genes in MG-63 cells [25]. Additionally, PD has been shown to inhibit proliferation and promote apoptosis in drug-resistant OS models [26,27] and enhances osteogenic differentiation in SAOS-2 and MG-63 cells, both alone and in combination with radiation therapy [28]. Notably, it has been reported that ROS-mediated endoplasmic reticulum stress and mitochondrial dysfunction can contribute to PD-induced apoptosis and cycle arrest in human nasopharyngeal carcinoma CNE cells [29]. Further evidence showed that PD inhibits glucose-6-phosphate dehydrogenase (G6PD), a rate-limiting enzyme in the pentose phosphate pathway, causing redox imbalance, which strongly affects cancer cell proliferation in vitro and in vivo in head and neck squamous cell carcinoma (HNSCC) [30]. Additionally, a recent study found that PD treatment synergistically enhances cisplatin’s antitumor effects in non-small cell lung cancer (NSCLC) cells by increasing ROS production through NADPH oxidase 5 (NOX5) activation [31].

Given these premises and considering the significant role of ROS in cancer, the present study aimed to investigate the anti-cancer effects of PD on human SAOS-2 and U2OS OS cell lines by examining associated changes in redox state, a previously unexplored aspect in OS. Specifically, using cell-based biochemical assays, we evaluated the inhibitory effects of PD on OS cell growth and normal osteoblasts (hFOB 1.19), focusing on ROS levels, GSH depletion, and intracellular levels of total iron and MDA as markers of ferroptosis [32,33]. The potential impact of PD-induced redox alterations on key metastatic properties, including adhesiveness and migratory capacity, was also investigated. Moreover, we assessed whether exposure to PD could affect the response of human OS cells to adaptive survival conditions during oxidative stress such as hypoxia and serum starvation, which are hallmarks of the tumor microenvironment [34,35]. Since previous studies have demonstrated that PD [36,37] as well as other polyphenolic pro-oxidant compounds [38,39] can regulate the level of Sirtuin 1 (SIRT1) and Nuclear factor erythroid 2-related factor 2 (Nrf2), which are key transcriptional masters governing oxidative stress response in both cancer and normal cells [40,41], we investigated the potential involvement of these proteins in PD-induced effects, by examining their protein and gene expression levels via Western blot and real-time PCR, following PD exposure. Finally, we explored the potential of PD as an adjuvant therapy for OS by evaluating its effect, in combination with the ROS-inducing chemotherapeutic agents DOX [42] and CIS [43,44], on the viability and proliferation of OS cells. The results presented herein support the use of PD in OS, revealing for the first time that PD-induced alteration in oxidative balance can contribute to its cytotoxic effects in this treatment-resistant, aggressive tumor.

## 2. Materials and Methods

### 2.1. Cell Culture and Treatments

The human OS cell lines SAOS-2 (HTB-85) and U2OS (HTB-96) and human normal osteoblast cell line hFOB1.19 (CRL-3602) were purchased from the American Type Culture Collection (Manassas, VA, USA). Cells were cultured in DMEM (Dulbecco’s modified Eagle’s medium) (4.5 g/L glucose)/Ham F12 (1:1) (Invitrogen, Carlsbad, CA, USA) and supplemented with 10% fetal bovine serum (FBS) (Euroclone s.p.a., Milano, Italy), Penicillin–Streptomycin Solution 100X (Gibco, Life Technologies, Carlsbad, CA, USA), and Amphotericin B 100X (Biowest, Riverside, MO, USA) at 37 °C in an atmosphere of 5% CO_2_. The culture medium was refreshed twice a week, and any non-adherent cells were removed during these routine changes. Experiments in serum-starved medium were performed in DMEM (4.5 g/L glucose)/Ham F12 (1:1) supplemented with 0.5% FBS. Polydatin (PD) (purchased from Sigma Aldrich, Milan, Italy) was dissolved at a concentration of 100 mM in 100% DMSO (Sigma Aldrich) as a stock solution and diluted with medium before each experiment. For treatments, cells were incubated for the indicated times in the presence of PD at various concentrations (dose range: 25–200 µM) or vehicle control (DMSO ≤ 0.1%). N-acetylcysteine (NAC) (Santa Cruz Biotechnology, Dallas, TX, USA) was added 1 h before PD administration to block ROS (10 mM NAC) and GSH depletion (1 mM NAC). Ferrostain-1 (Fer-1) (Sigma Aldrich), 1 µM, was used as a ferroptosis inhibitor. For experiments involving doxorubicin (DOX) and cisplatin (CIS) (both purchased from Enzo Life Sciences, Inc. Farmingdale, NY), SAOS-2 and U2OS cells were incubated in the presence or absence of PD for the indicated times and at the indicated doses of PD or DMSO. DOX was dissolved in 100% DMSO and CIS in 0.9% NaCl.

### 2.2. Cell Viability

Cell viability was determined by MTT 3- (4,5-dimethylthiazolyl-2)-2,5-diphenyltetrazolium bromide) assay (Merk Life Science s.r.l., Milano, Italy), as previously described [6,45]. SAOS-2, U2OS, and hFOB 1.19 line cells were seeded at 5 × 10^3^/well in 96-well plates and incubated in the growth medium at 37 °C to allow cell attachment. After 24 h, the medium was changed and cells were incubated for 24, 48, and 72 h with increasing doses of PD (0–200 µM) or with DMSO (≤0.1%). At the end of each treatment period, the cells were incubated for a further 2 h at 37 °C with 5% CO_2_ in 20 µL MTT solution (5 mg/mL in PBS w/o Ca^2+^ and Mg^2^, Merk Life Science s.r.l., Milano, Italy). Then, a volume of 100 µL/well of extraction buffer (5% SDS in N,N-Dimethylformamide) was added, followed by 2 h of incubation at 37 °C with 5% CO_2_ to solubilize the formazan crystals prior to measuring the absorbance with a microplate reader at 570 nm, using a multi-plate TECAN Spark R reader (Tecan Group Ltd., Männedorf, Switzerland). The percentage survival of the cultures treated with PD was calculated by normalization of their O.D. values to those of the control cultures treated with DMSO. IC50 values were calculated using GraphPad Prism 9.01 software (San Diego, CA, USA).

### 2.3. Proliferation Assay

For the proliferation assays, SAOS-2, U2OS, and hFOB 1.19 were seeded in 24-well plates, either pre-coated with rat tail collagen I (50 µg/mL) at a density of 6 × 10³ cells/well, or on non-coated plates at 1 × 10³ cells/well. After incubation in growth medium at 37 °C to allow for cell attachment, the cells were treated with PD (25–200 µM) or vehicle control (DMSO ≤ 0.1%) for 24, 48, and 72 h. Following treatment, cell proliferation was assessed by measuring live cell confluence using the confluence function of the Spark microplate reader (Tecan Group Ltd., Männedorf, Switzerland) [46]. The percentage of live cell confluence was calculated by normalizing the values of PD-treated cells to those of the DMSO control cultures.

### 2.4. Wound-Healing Assays

Cell migration was examined using the wound-healing assay, as previously reported [45,46]. Briefly, SAOS-2 and U2OS cells were seeded at a density of 1 × 10^4^ cells per well in conventional culturing 96-well plates until full confluence. A scratch was created in the cell monolayer using a micropipette tip and debris was removed by washing with PBS. Cells were further incubated at 37 °C with 5% CO_2_ and serum-free medium containing PD (25 and 50 µM) or vehicle control (DMSO ≤ 0.05%), and allowed to migrate for up to 48 h. Images were captured at different time points after the scratch (0–48 h) using the Tecan Spark instruments (Tecan Group). The images acquired were analyzed quantitatively, using a specific wound-healing tool of ImageJ software (Bethesda, MD, USA). The percentage of relative wound closure was calculated using the following equation, as previously described [46]: % wound closure = (area of the wound at time 0 h) − (area of the wound at time 24 or 48 h)/(area of the wound at time 0 h) × 100.

### 2.5. Adhesion Assay

To assess the effect of PD on cell adhesion, SAOS-2 and U2OS cells (2 × 10^3^ cells/well), were seeded on 24-well plates pre-coated with rat tail collagen I (50 µg/mL), in the presence of either PD (25 and 50 µM) or vehicle control (DMSO ≤ 0.05%). After 24 h of plating, non-adherent cells were removed by washing the wells twice with PBS, and adherent cells were stained with 0.5% crystal violet dissolved in a 20% *v*/*v* methanol solution at 37 °C (Sigma Chemical Co., St. Louis, MO, USA). Following washes to remove excess stain, images were captured using an Olympus CKX53 inverted microscope equipped with an EP50 Microscope Digital Camera (Olympus Life Science, Waltham, MA, USA) and attached cells were counted. Attachment data are expressed as a percentage of the number of adherent cells in PD-treated samples compared to DMSO-treated controls.

### 2.6. Detachment Assay

SAOS-2 and U2OS cells were seeded and treated as described in the adhesion assay. Following 24 h of incubation, initial images were captured. Cells were then detached by incubating the plates for 1 h at 37 °C with vigorous shaking (240 rpm). The remaining attached cells were stained with crystal violet as described above, and images were captured. Cell detachment data are expressed as the percentage of the number of remaining attached cells vs. the initial cell number.

### 2.7. ROS Detection Assay

Intracellular ROS formation was assessed using a DCFDA/H2DCFDA cellular ROS detection assay (ab113851, Abcam, Cambridge, UK), in accordance with the manufacturer’s specifications. Briefly, SAOS-2, U2OS, and hFOB 1.19 cells were seeded in a 96-well plate (2 × 10^4^ cells/well) and cultured for a day to allow cell adhesions. The following day, the medium was replaced with phenol red-free medium and cells were treated with PD (100 or 200 µM) or vehicle control (DMSO ≤ 0.1%) with or without pre-treatment with NAC (10 mM for 1 h prior to PD exposure). After 24 h, cells were washed with 100 µL/well of 1× buffer and stained with the H2DCFDA probe at 37 °C with 5% CO_2_ in the dark, according to the manufacturer’s instructions. Fluorescence was measured at an excitation/emission wavelength of 485/538 nm using the Infinite^®^200 PRO multi-well plate reader (Tecan Group Ltd., Männedorf, Switzerland). The amount of ROS produced was quantified based on the fluorescence intensity emitted. ROS levels are expressed as a percentage of DCFDA fluorescence intensity relative to DMSO-treated cells.

### 2.8. Glutathione (GSH) Assay

The GSH/GSSG Ratio Detection Assay Kit II (Abcam, ab205811, Cambridge, UK) was used to detect reduced (GSH) and oxidized (GSSG) glutathione levels, following the manufacturer’s protocol. SAOS-2, U2OS, and hFOB 1.19 cells were seeded at a density of 2.5 × 10^4^ cells/well in a 24-well plate and incubated for 24 h. Cells were subsequently treated with either 100 or 200 µM of PD or a vehicle control (DMSO ≤ 0.1%), with or without pre-treatment with NAC (1 mM, administered 1 h prior to PD exposure) and then cultured for an additional 24 h. After completion of the experiment, medium was removed and cells were washed with cold PBS. Cells were lysed and homogenized for 15 min, and clear supernatant was collected and transferred to a new tube. For GSH detection, 50 µL of GSH assay mixture was added to standards and samples. For GSSG measurement, 50 µL of total glutathione assay mixture was added to standards and samples. After incubation of 20 min, fluorescence was measured at Ex/Em = 490/520 nm using the Infinite^®^200 PRO multi-well plate reader.

### 2.9. Measurement of Iron Level

The Iron Assay Kit (Abcam, ab83366, Cambridge, UK) was used to measure total iron levels in lysates of SAOS-2 and U2OS cells treated with PD at IC50 doses or DMSO (≤0.1%), following the manufacturer’s instructions and previously described methods [47,48]. Briefly, 50 μL of lysates of equal protein concentration (determined by protein assay—Bio-Rad, Hercules, CA, USA) was added to a 96-well plate after lysis of 3 × 10^5^ cells in 300 μL of lysis buffer provided by the kit. Calibration standards were included in the same plate. An iron reductant was added to the wells to convert all ferric iron to ferrous iron for the determination of total iron (II + III). The plate was placed in a 37 °C incubator for 30 min before the Iron Probe was added and incubated for a further 60 min. Unbound iron (II) interacts with the Iron Probe to form a stable-colored compound, which was immediately assayed using a colorimetric microplate reader (OD = 593 nm). The standard curve (provided with the kit) allows the direct determination of iron (II) and total iron content in the test samples. Total iron–iron (II) was used for the calculation of iron (III). Values are expressed as nmoles present in the 50 µL cellular lysates.

### 2.10. Lipid Peroxidation (MDA) Assay

The intracellular MDA concentration in cell lysates or tissues was assessed using a lipid peroxidation assay kit (cat. no. ab118970, Abcam) according to the manufacturer’s instructions. The reaction of MDA in the samples with thiobarbituric acid (TBA) resulted in the generation of an MDA-TBA adduct. The MDA-TBA adduct was quantified fluorometrically at an excitation/emission wavelength of 532/553 nm using the Infinite^®^200 PRO multi-well plate reader. The experiments were repeated three times for each group.

### 2.11. Quantitative RT-PCR Analysis

To assess the effects of PD treatment on gene expression, SAOS-2 and U2OS cells were treated with PD at its IC50 dose for 24 and 48 h. Cells were then collected and prepared for analysis of the target genes SIRT1 and Nrf2. Total RNA was extracted using TRIZOL Reagent (Roche Diagnostics GmbH, Mannheim, Germany). The quality of the extracted RNA was assessed by measuring the absorbance ratio at 260/280 nm using the NanoQuant Plate with an Infinite^®^200 PRO multi-well plate reader (Tecan Group Ltd., Männedorf, Switzerland). The RNA was reverse-transcribed into cDNA using the SensiFAST™ cDNA Synthesis Kit (Bioline, Meridian Bioscience, London, United Kingdom) following the manufacturer’s instructions. Gene expression levels were quantified using iTaq Universal SYBR Green Supermix (Bio-Rad Laboratories, Hercules, CA, USA), and quantitative real-time PCR (qRT-PCR) was performed using a LightCycler 96 Real-Time PCR System (Roche Diagnostics GmbH). For data analysis, the expression of target genes was normalized using the ΔΔCt method, with human glyceraldehyde 3-phosphate dehydrogenase (GAPDH) as the reference gene. Relative quantification was carried out using LightCycler^®^96 system software version 1.1 (Roche). The primer sequences were as previously designed [6] and synthesized by Merck (Life Sciences, Milano, Italy). The Q-PCR primers for SIRT-1 were 5′-TAGCCTTGTCAGATAAGGAAGGA-3′ and 5′-ACAGCTTCACAGTCAACTTTGT-3′. The Q-PCR primers for Nrf2 were 5′-TGAGGTTTCTTCGGCTACGTT-3′ and 5′-CTTCTGTCAGTTTGGCTTCTGG-3′.

### 2.12. Western Blot

After 24 or 48 h of treatment with PD at its IC50 or DMSO, cells were harvested, washed twice with cold PBS, and lysed in RIPA buffer containing 150 nM NaCl, 50 mM Tris-HCl, 0.1% SDS, 1% Triton X-100, 1% Na cholate, 1% NaOV, 1 mM NaF, 1 mM EDTA, PMSF 1X, and a Protease Inhibitor Cocktail (Cell Signaling Technology, Danvers, MA, USA). For immunoblotting analysis, from 30 to 60 µg of cell lysates was resolved in precast gels (Mini-PROTEAN^®^ TGX™ Precast Gels 4–20%, BioRad, Hercules, CA, USA) and then transferred to a PVDF membrane (Amersham, Buckinghamshire, UK). The membrane was blocked with EveryBlot Blocking Buffer (BioRad, Hercules, CA, USA) and probed with specific primary and secondary antibodies. The blots were developed by using Enhanced Chemiluminescence (ECL) detection systems (Amersham, UK). The following primary antibodies were used: GAPDH (dilution 1:10,000) (GeneText Irvine, Irvine, CA, USA, GTX100118) as a control, SIRT1 (B-10) (dilution 1:100) (sc-74505 Santa Cruz Biotechnolgy Inc. CA, USA), and Nrf2 (A-10) (dilution 1:100) (sc-365949 Santa Cruz Biotechnolgy Inc. CA, USA). The intensities of bands were observed in Azure c280 (Aurogene S.r.l., Roma, Italy) and measured through an image analysis program (ImageJ (1.54m), public domain software; NIH, Bethesda, MD, USA) and quantified using a scale of arbitrary units (AU). The results obtained were analyzed using statistical software (Prism 9 ver. 9.0.0 (121), GraphPad Software).

### 2.13. Proliferation Assay Under Hypoxic Conditions

OS cells were seeded into two 96-well plates at a density of 8 × 10^3^ cells/well and allowed to adhere for 24 h. The following day, cells were treated with PD (100 and 200 µM) or vehicle control (DMSO ≤ 0.1%) for 3 h under normoxic conditions. After treatment, the cells were subjected to two different culture conditions. For hypoxic conditions, one plate was placed inside the patented Humidity Cassette and incubated in the Infinite^®^200 PRO multi-well plate reader (Tecan Group Ltd., Männedorf, Switzerland), utilizing the gas and humidity control function (37 °C, 5% CO_2_, 3% O_2_) for 24 h. The second plate, for normoxic culture, was incubated for an additional 24 h in a standard incubator. Cell confluence was measured at the 24-h mark using the Infinite^®^200 PRO reader to assess the proliferation rate of cells in each group.

### 2.14. Combined PD and Chemotherapeutic Agent Treatment

The cytotoxic effects of the chemotherapeutic agents DOX and CIS on SAOS-2 and U2OS cells were initially assessed using the MTT assay to determine the IC_50_ of each drug. SAOS-2 and U2OS cells were seeded at a density of 5 × 10^3^ cells per well in 96-well plates and incubated in growth medium at 37 °C to allow for cell attachment. After 24 h, the cells were treated with increasing concentrations of DOX (0.5–25 µM) or CIS (1.5–60 µM), along with their respective vehicle controls, DMSO or 0.9% NaCl, for 48 h. The IC50 values were then determined for both agents and used in subsequent combination treatments with PD. For combined PD–DOX and PD–CIS treatments, cells were treated with PD (100 µM, 200 µM or IC50 doses) or vehicle controls in combination with DOX (IC50: 10 µM) or CIS (IC50: 20 µM) for 48 h, with or without pre-treatment with NAC (10 mM, administered 1 h prior to drugs exposure). The cytotoxic effects of the combined PD–DOX and PD–CIS treatments were evaluated using both the MTT assay and proliferation assay on non-coated plates, as described above. All experiments were conducted in triplicate and repeated three times. IC50 values were calculated using GraphPad software (Prism 9 ver. 9.0.0 (121)).

### 2.15. Statistical Analysis

All data are expressed as the mean ± S.D. Statistical analyses were performed using an unpaired *t*-test to compare two groups, and one-way ANOVA followed by Tukey’s post hoc test for comparisons involving three or more groups (GraphPad Prism version 9.0.0, software version 9.0.0). In all studies, the significance threshold was set at *p* ≤ 0.05. Experiments were independently performed at triplicated or more. Representative experiments are shown.

## 3. Results

### 3.1. Cytotoxic Effects of PD on Human OS Cells

The MTT and proliferation assays were initially conducted to evaluate the sensitivity of the human OS cell lines, SAOS-2 and U2OS, and normal osteoblasts, hFOB 1.19, to PD treatment. Proliferation assays were performed on both type-I-collagen-coated and uncoated plates to assess the potential role of extracellular matrix proteins, such as collagen, in the PD-induced cytotoxic process. Cultured cells were exposed to increasing concentrations of PD (25–200 µM) or DMSO as vehicle control up to 72 h, prior to the MTT or proliferation experiments. The results reported in Figure 1A show that incubating SAOS-2 and U2OS cells with PD led to a noticeable trend of reduced cell viability in both OS cell models, which aligns with previous studies [27,28].

In particular, after 48 h of treatment, the effect of PD achieved statistical significance at concentrations ≥ 50µM with the half-maximal inhibitory concentration (IC50) determined to be approximately 117 µM for SAOS-2 cells and 160 µM for U2OS cells.

Consistent with the MTT results, proliferation experiments on coated surfaces revealed a time-/dose-dependent suppressive effect on both SAOS-2 and U2OS cells exposed to PD, with a marked decrease especially at the highest tested doses (100–200 μM) (*p*-value range: 0.01 to 0.001) (Figure 1B). Importantly, cell confluence analysis on uncoated plates under identical conditions yielded similar results, indicating that the presence of collagen does not significantly influence PD-induced cytotoxicity in OS cells (Appendix A). Notably, treatment of non-tumorigenic cells (hFOB 1.19) at the selected concentrations and times had a negligible effect on cell survival and proliferation, with the latter dropping below 80% only at the highest dose tested after 72 h of treatment, indicating that PD exhibited selective cytotoxic effects towards malignant cells (Figure 1A,B).

### 3.2. Effect of PD on the Migration and Adhesion Abilities of OS Cells

Metastasis relies on tumor cells’ ability to adhere to extracellular matrix proteins and survive at distant sites. OS is known for its high metastatic potential, which correlates with poor prognosis and unfavorable therapeutic outcomes [3,49,50]. We investigated whether PD effectively blocks key hallmarks of metastatic behavior in vitro, such as migration, adhesion, and detachment. Non-toxic (25 µM) and low-toxic (50 µM) doses of PD were used to minimize interference from cell toxicity. The anti-migratory effect of PD on OS cells was assessed by wound-healing assay experiments for up to 48 h. As shown in Figure 2A, both SAOS-2 and U2OS cells displayed a diminished migratory ability at both tested concentrations. Notably, after 48 h of treatment with 50 μM PD, the percentage of wound closure was ∼18% for SAOS-2 cells and ∼20% for U2OS cells, compared to ∼38% and ∼45% in the respective control groups (*p* < 0.001 and *p* < 0.01, for SAOS-2 and U2OS, respectively) (Figure 2A).

Our data clearly show that treatment with PD caused a significant inhibition of cell migration even at subtoxic concentration in OS cells. Figure 2B,C presents the results of cell adhesion and detachment assays, respectively, performed after 24 h of incubation on type I collagen-coated plates. The adhesive capacity of SAOS-2 and U2OS cells was evaluated by comparing the number of attached cells following PD administration (25 and 50 µM; 24 h) with that of the control group. The results revealed a dose-dependent inhibition of cell adhesion capability (∼45% and ∼35% with 25 and 50 μM PD, respectively, for SAOS-2 cells, and ∼32% and ∼16% with 25 and 50 μM PD, respectively, for U2OS cells) compared to DMSO controls, (*p* < 0.001) (Figure 2B). Figure 2B, left panel, illustrates representative microscopical images of cell adhesion assays. The anti-metastatic potential of PD against SAOS-2 and U2OS cells was further investigated by detachment assay. Following 24 h of incubation on pre-coated plates, the cells were detached, and the number of remaining attached cells was compared between PD-treated and DMSO-treated samples. As shown in Figure 2C, the percentage of detached cells increased in PD-treated SAOS-2 groups to approximately 61% and 65% at 25 and 50 μM PD, respectively, and to around 53% and 78% in PD-treated U2OS groups at 25 and 50 μM PD, respectively, compared with the control group (37% and 18% for SAOS-2 and U2OS, respectively; *p*-value range: 0.01 to 0.001) (Figure 2C). Microscopic pictures from the cell detachment assay following 24 h of treatment with 25 and 50 µM PD are shown in Figure 2C, left panel.

These results indicate that PD not only partially impairs the cells’ ability to establish adhesion, but also weakens the bonds that maintain their attachment to the surface. Overall, based on the above results, PD demonstrates promising dual cytotoxic and anti-metastatic properties in SAOS-2 and U2OS cells.

### 3.3. PD-Induced Modulation of Intracellular Levels of ROS and GSH

The anti-cancer effects of certain plant metabolites have been shown to involve ROS generation [51,52]. In this regard, it has been reported that contrary to its antioxidant properties, PD can act as a pro-oxidant in certain cancer cells by causing redox imbalance, which results in endoplasmic reticulum stress, cell cycle arrest, and apoptosis [29,30].

To assess the potential role of intracellular ROS in the regulatory mechanisms leading to PD-induced anti-OS activity, SAOS-2 and U2OS cells, along with normal human osteoblasts (hFOB 1.19), were incubated for 24 h at 100 μM and 200 μM PD, corresponding to concentrations below and above the IC50 for SAOS-2 and U2OS. Cells were either pre-treated with the ROS scavenger N-acetyl-L-cysteine (NAC) or left untreated. Intracellular ROS levels were subsequently measured using a DCFDA/H2DCFDA-based assay.

As reported in Figure 3A, PD promoted intracellular ROS generation by approximately 2- and 3-fold in SAOS-2 and U2OS, respectively, compared to DMSO-treated cells, arbitrarily set as 100% (*p* < 0.01 and *p* < 0.001 for SAOS-2 and U2OS respectively). In contrast, PD exposure did not lead to an increase in ROS levels in non-tumorigenic hFOB 1.19 cells, suggesting that PD selectively targets cancer cells (Appendix A).

Pre-treatment of OS cells with NAC 1 h prior to PD treatment resulted in a significant decrease in ROS generation in both OS cell lines, compared to cells treated with PD alone (100 and 200 µM) (p-value range: 0.05 to 0.01) (Figure 3A). Interestingly, the increase in ROS levels in SAOS-2 and U2OS cells after PD administration was detected only at a cytotoxic dose range (100–200 µM). ROS generation was not observed at concentrations below 100 µM, suggesting a link between pro-oxidant stimulation by PD and inhibition of cell growth (Appendix A).

GSH metabolism is tightly regulated and has also been implicated in redox signaling [53]. The balance between oxidants and antioxidants, in particular the GSH/oxidized GSH (GSSG) ratio, is a useful measure of cellular redox status [53,54]. To assess whether the PD-induced ROS generation was accompanied by a reduction in the GSH/GSSG redox couple, we measured the intracellular GSH and GSSG levels using a fluorimetric method, following 24 h of PD stimulation at cytotoxic doses of 100 and 200 µM. As a result, PD administration led to a significantly decreased percentage of the GSH/GSSG ratio in both SAOS-2 and U2OS cells compared with DMSO-treated groups (by ∼0.7-fold for SAOS-2 cells and ∼0.6-fold for U2OS cells, *p* < 0.01), irrespective of PD concentrations (Figure 3B). This suggests a potentially diminished ability of PD-treated OS cells to maintain intracellular redox homeostasis. Cells likely consumed their cellular antioxidant stores to counteract the PD-induced ROS accumulation, leading to GSH depletion.

Consistently, pre-incubation with NAC partially abrogated the GSH depletion induced by PD in SAOS-2 and U2OS cells (*p* < 0.05) (Figure 3B). With regard to hFOB 1.19 cells, no significant changes were observed in GSH and GSSG levels after 24 h of PD exposure (Appendix A). Notably, unlike OS cells, treatment with NAC alone led to a marked increase in the GSH/GSSG ratio compared with the control counterpart, probably due to the NAC-induced enhancement of the intracellular cysteine pool, resulting in increased glutathione (GSH) levels [55]. Given the above results, we investigated whether ROS were involved in PD-induced anti-proliferative effects in SAOS-2 and U2OS cells. For this purpose, OS cells were pre-incubated with NAC and assessed for viability using the MTT assay, and a proliferation assay on pre-coated wells, following 48 h of PD exposure (100 and 200 µM). NAC significantly attenuated PD-induced growth inhibition in SAOS-2 and U2OS cells (*p*-value range: 0.05 to 0.001) (Figure 3C,D), strongly suggesting a causal link between the early induction of ROS and PD-induced cell toxicity in OS cells. Taken together, these observations indicate that PD, at doses near the growth IC50 values, influenced cellular redox status in SAOS-2 and U2OS without side effects on normal osteoblasts.

### 3.4. PD-Induced Effect on Iron Accumulation and Lipid Peroxidation in OS Cells

Recent evidence indicates that oxidative stress and ferroptosis, an iron-dependent form of cell death, share overlapping molecular events including glutathione depletion, impaired redox potential, and ROS increment with concomitant lipid peroxidation [32,33].

Ferroptosis has been shown to influence the progression of OS, suggesting its potential therapeutic value [56,57]. Given that a variety of drugs and biologically active molecules have been found to induce ROS-mediated OS cell death through ferroptosis [48,58,59], we investigated whether the PD-induced change in the oxidative balance in SAOS-2 and U2OS cells could be related to this type of programmed cell death. The ferroptosis-related indicators, iron and MDA levels, were assessed in SAOS-2 and U2OS cells treated with PD at their respective IC50 concentrations (120 µM for SAOS-2 and 160 µM for U2OS) for 24 h. Figure 4A shows the intracellular levels of total iron and its oxidized (Fe(II)) and reduced (Fe(III)) forms after treatments. The data indicate a slight increase in total and ferrous iron levels in both OS cell lines treated with PD, compared to the control counterpart (*p* < 0.01). Additionally, MDA levels were elevated in both U2OS and SAOS-2 cells following PD treatment (*p* < 0.05) (Figure 4B).

Based on these results, we next investigated the potential involvement of ferroptosis in PD-induced anti-OS activity. SAOS-2 and U2OS cells were treated with PD (IC50) for up to 48 h, with or without the ferroptosis inhibitor Ferrostatin-1 (Fer-1, 1 µM). Cell viability was then measured using the MTT assay. Interestingly, Fer-1 treatment significantly prevented PD-induced cell death in both SAOS-2 and U2OS cells (Figure 4C), with a particularly strong effect in U2OS (*p* < 0.001), suggesting that PD may promote cytotoxic effects via ferroptosis induction.

### 3.5. Detection of Oxidative Stress-Related Gene Expression and Protein Level After PD Treatment

Since our analysis showed distinctive features of increased oxidative stress in PD-treated cells, we investigated whether PD stimulation was accompanied by changes in gene expression and protein level of oxidative stress-related biomarkers, such as SIRT1 [60,61] and Nrf2 [62,63], which have also been proposed as potential targets for PD in various pathological conditions associated with redox imbalance [37,64]. Nrf2 has also emerged as a key regulator of iron metabolism in the process of ferroptosis, especially in the context of OS [56,65]. Protein and gene expression levels of SIRT1 and Nrf2 were analyzed by Western blot analysis and real-time PCR, respectively, at 24 and 48 h after the addition of IC50 doses of PD. As shown in Figure 5A, SIRT1 and Nrf2 expression significantly increased in SAOS-2 and U2OS cells after 24 h of PD treatment (*p*-value range: 0.05 to 0.01). However, this upregulation was transient, returning to baseline levels within 48 h. While the initial increase in SIRT1 and Nrf2 mRNA expression suggests an early antioxidant response to PD-induced ROS elevation, the lack of changes in total protein levels at both 24 and 48 h, as observed in immunoblot analysis (Figure 5B), indicates an insufficient compensatory mechanism. This inadequate antioxidant response may contribute to PD-induced cytotoxicity.

### 3.6. Effect of PD Treatment on the Sensitivity of OS Cells to Hypoxic Conditions and Serum Starvation

Many solid tumors including OS grow so quickly that they often exceed their own vascular supply to sequentially lead to local low oxygen tension (hypoxia) and nutritional deficiency [34,66]. This harsh environment forces tumor cells to adapt, especially to the low oxygen levels, leading to more aggressive and diverse tumor cell populations that are resistant to multiple therapies [67,68]. In our study, we investigated whether exposure to PD could influence the response of human OS cells under hypoxia and serum starvation, which are key features of the tumor microenvironment. Figure 6A shows the effect of PD administration on the proliferation efficiency of OS cells under hypoxic conditions compared to normoxic conditions.

SAOS-2 and U2OS cells were exposed to either normoxia (21% O_2_; 24 h) or physiological hypoxia (3% O_2_; 24 h) using the gas and humidity control function of the Spark^®^ Microplate readers (TECAN), in the absence or presence of PD (100 and 200 µM). Cell live confluence after 24 h was measured to compare the growth efficiency of the cells in each group. As shown in Figure 6A, hypoxia induced a significant increase in the proliferation rate of untreated SAOS-2 and U2OS cells compared to normoxic conditions (*p* < 0.01 and *p* < 0.001 in SAOS-2 and U2OS, respectively), consistent with previous research findings [67,68]. Interestingly, PD administration inhibited the growth of hypoxic cell groups after 24 h of incubation, possibly predisposing the malignant cells to a less aggressive phenotype. In particular, in SAOS-2 cells, this effect gained statistical significance at the highest concentration of PD, 200 µM (*p* < 0.05). Conversely, in U2OS cells, a significant decrease in cell confluence was observed at both tested doses (*p* < 0.001).

The effect of PD was also investigated under starvation conditions. Specifically, the effect of serum-reduced medium (0.5% FBS) on OS cells was investigated by assessing the cell viability after PD administration (100 and 200 µM) up to 72 (Figure 6B). Results from the MTT revealed that SAOS-2 and U2OS cells treated with cytotoxic doses of PD showed increased sensitivity to serum starvation compared to untreated cells (*p* < 0.01), as evidenced by a growth delay over time.

Collectively, the above data indicate that PD is able to sensitize SAOS-2 and U2OS cells even under conditions like hypoxia and serum starvation where cancer cells typically adapt to oxidative stress and enhance their survival mechanisms against anti-cancer agents [66,69].

### 3.7. Effects of the Combined Treatment of PD and the Chemotherapeutic Agents, DOX and CIS, on OS Cells Growth and Viability

To date, numerous studies have demonstrated the protective role of plant-derived bioactive compounds, including PD, when combined with therapeutic agents [26,31,70,71]. These combinations often promote synergistic effects, mitigating the side effects of chemotherapy and enhancing therapeutic efficacy [11]. To further investigate the cytotoxic effects of PD treatment on SAOS-2 and U2OS cells, we evaluated the efficacy of PD in combination with the chemotherapeutic agents, DOX and CIS, which are currently used in the therapeutic regimen of OS [4,5]. Both DOX and CIS are known to trigger metabolic alterations in cancer models, which can result in secondary detrimental effects like endogenous ROS production and ferroptosis [42,43,44]. SAOS-2 and U2OS cells were preliminarily treated with increasing concentrations of DOX and CIS for 48 h to determine the IC50 for each drug. MTT assays confirmed that both drugs, individually administered, reduced cell viability in a dose-dependent manner (Figure 7A,B). DOX exhibited a more pronounced cytotoxic effect than CIS, with approximate IC50 values of 10 µM and 20 µM, respectively, which aligns with previous findings [70]. To determine whether PD could sensitize OS cells to the chemotherapeutic agents, cells were treated with PD (100 or 200 µM) in combination with DOX or CIS at their respective IC50 concentrations for 48 h.

Figure 7C,E presents the combined effects of PD and either DOX or CIS on cell viability (MTT assay). Cell proliferation on non-coated plates was also investigated and is reported in Appendix A. The combination treatments significantly reduced cell survival and proliferation compared to single-agent treatments in both cell lines. The PD-DOX combination demonstrated enhanced efficacy compared to PD-CIS. Specifically, the combination PD (100 µM)-DOX reduced cell viability by approximately 35% and 39% in SAOS-2 and U2OS cells, respectively (*p* < 0.01 and *p* < 0.05), compared to the DOX_IC50_ dose. A more pronounced effect was observed with the PD (200 µM)-DOX combination, leading to around 27% and 20% viability reductions in SAOS-2 and U2OS cells, respectively (*p* < 0.001). These results were corroborated by live cell confluence measurements, which showed similar trends (Appendix A).

In contrast, the PD (100 µM)-CIS combination exhibited minimal, non-significant effects on cell growth and proliferation in SAOS-2 cells, while limited effects were observed in U2OS cells (approximately 43% reduction), compared to the CIS_IC50_ dose (*p* < 0.05) (Figure 7E and Appendix A). However, the PD (200 µM)-CIS combination led to moderate reductions in cell viability and proliferation in both cell lines (by approximately 35% compared to CIS alone (*p* < 0.05 and *p* < 0.01 for SAOS-2 and U2OS, respectively). Notably, the inhibitory effects of combined PD and DOX/CIS treatment on cell growth, when administered at their respective IC50 doses for 48 h, were markedly attenuated by pre-treatment with the ROS scavenger NAC (10 mM, 1 h prior to treatment) (*p*-value range: 0.05 to 0.0001) (Figure 7D,F). This suggests that ROS may play a critical role in mediating the reduction in OS cell viability observed with combination treatment.

## 4. Discussion

OS is a complex disease arising from the intricate interplay of various factors that disrupt bone tissue homeostasis by deregulating cellular signaling pathways [1,3]. Despite being relatively rare, OS is the most common primary bone cancer in children and young adults. The lack of specific diagnostic and prognostic markers, coupled with the heterogeneity of OS subtypes, hinders the development of effective therapies [4,5]. To address these challenges, there is a growing interest in exploring natural non-toxic substances as potential anti-cancer agents [11,12]. Traditional Chinese medicine offers a rich source of bioactive molecules for drug discovery [12,72].

PD, a stilbenoid compound abundant in various fruits and vegetables, stands out as a versatile therapeutic agent able to modulate many biological processes and to change a large variety of phenotypes at cellular and organismal levels [17,21]. Its structure, characterized by a conjugated system with phenolic groups, endows it with potent antioxidant properties linked to chemopreventive and anti-inflammatory effects [18,73]. However, the biological effects of PD are multifaceted, ranging from antioxidant to pro-oxidant, depending on factors such as concentration, cellular environment, and experimental conditions [18,21]. This dual nature highlights the complexity of PD’s pharmacological profile. In our study, we present the first evidence that cytotoxic doses of PD alter the redox state in human SAOS-2 and U2OS cells. This redox imbalance renders these cells more vulnerable to challenging conditions where cancer cells typically adapt to survive (e.g., hypoxia, nutrient deprivation, and chemotherapy treatment). PD significantly impedes cell growth in both lines by inducing ROS generation and depleting GSH. Additionally, we observed an increase in intracellular total iron and its oxidized form, Fe(II), along with elevated MDA concentrations, suggesting a potential involvement of ferroptosis in inhibiting OS cell proliferation. PD’s ability to suppress metastatic properties, such as cell adhesion and migration, appears to be independent of its pro-oxidant activity.

Oxidative stress is increasingly recognized as a valuable target for cancer therapy [7,8,9]. Mounting evidence links elevated oxidative stress to cancer progression inhibition, while high GSH levels are associated with tumor growth and chemotherapy resistance [54]. To exploit this vulnerability, novel therapeutic strategies have been developed involving either elevating ROS levels or inhibiting the cell’s endogenous antioxidant defense mechanisms, aiming to enhance treatment efficacy and overcome drug resistance [6,7].

PD-induced ROS generation, iron increase, and reduction in the GSH/GSSG redox couple indicates impaired redox homeostasis in SAOS-2 and U2OS cells. To counter oxidative stress, cells likely deplete their antioxidant reserves, including GSH, highlighting PD’s potential as an anti-OS agent through redox imbalance. NAC partially reversed PD-induced ROS elevation and GSH depletion, supporting ROS as a key mediator of PD’s effects. Interestingly, ROS generation occurred only at cytotoxic PD concentrations (100–200 µM), suggesting a correlation between pro-oxidant stimulation by PD and cell growth inhibition. Beyond its anti-proliferative effects, PD also influences key metastatic processes by suppressing cell motility and adhesion while promoting the detachment of SAOS-2 and U2OS cells, even at subtoxic doses, which minimize interference from cell toxicity.

This highlights PD’s significant impact in both preventing cells from establishing adhesion on a coated surface and weakening the bonds that keep them attached. Although ROS are implicated in processes like metastasis and angiogenesis [74,75], the roles of specific ROS subtypes and the underlying regulatory mechanisms remain unclear due to inconsistent findings across studies, which may be attributed to differences in ROS type, ROS inducers’ dosage, and cell type. Notably, our study suggests that PD’s anti-metastatic effects are not directly linked to its pro-oxidant activity, as no increase in ROS levels was observed under conditions where PD effectively suppressed the metastatic behavior of OS cells.

To fully understand PD-mediated cell death involving ROS accumulation, identifying the ROS source and the upstream signaling pathways is crucial. While PD can exert antioxidant effects in several anti-inflammation and antioxidant models (e.g., suppressing ROS generation via NADPH oxidase inhibition [76], inhibition of the TLR4/NF-kB p65 [76], or activating the Nrf2/HO-1 pathway [77]), its role in inducing ROS and promoting cell death in cancer cells is less defined. Notably, the relationship between PD and ROS appears cell type-specific. Only a few studies have demonstrated the involvement of ROS in PD-induced cytotoxicity in cancer models [30,31]. For example, in HNSCC, PD inhibits G6PD, the key enzyme in the pentose phosphate pathway, leading to ROS accumulation, ER stress, and cell death [30]. In nasopharyngeal carcinoma, PD triggers redox imbalance, activating the unfolded protein response (UPR) and disrupting mitochondrial function [29]. Additionally, PD enhances cisplatin’s antitumor activity in NSCLC by inducing NOX5 expression and promoting ROS generation [31]. Importantly, both our findings and the above-mentioned studies show minimal ROS generation and limited effects on non-tumorigenic cells, suggesting PD’s selectivity for cancer cells. Understanding the differential response to PD between normal and malignant cells warrants further investigation. Although PD has been associated with increased ROS levels and tumor cell suppression, its potential to induce ferroptosis in cancer models remains unexplored.

Ferroptosis, an iron-dependent form of oxidative cell death driven by GSH depletion and GPX4 inactivation, is a promising therapeutic target for cancer, including OS [33,57]. Several plant-derived compounds, such as EF24, a curcumin derivative [48], and Bavachin [58], have shown the ability to inhibit OS progression by triggering ferroptosis, which can coexist and cooperate with other cell death pathways in OS cells.

Our findings indicate that ferroptosis may contribute to PD-induced redox imbalance in SAOS-2 and U2OS cells, as evidenced by elevated cytosolic iron levels and partial reversal of PD-induced cytotoxicity by the ferroptosis inhibitor Fer-1. While further research is required to clarify the underlying mechanisms, these results point to a possible link between PD-mediated OS suppression and ferroptosis in these cancer cells.

SIRT1 and Nrf2 are key regulators of the cellular antioxidant response, working together to protect against oxidative damage [40,41]. In OS, SIRT1 exhibits both tumor-suppressive and oncogenic roles [60,78], while Nrf2 is linked to OS progression and activation of OS-supportive anabolic metabolism [79]. Moreover, inhibiting Nrf2 can reverse ferroptosis resistance in MG63 and SAOS-2 OS cells treated with GPX4 inhibitors [65]. Both SIRT1 [36,37] and Nrf2 [64,80] have been implicated as potential targets for PD in various redox-imbalanced conditions. Based on these premises, we investigated their potential involvement in PD-induced ROS-driven cytotoxicity and found that PD treatment led to a transient upregulation of SIRT1 and Nrf2 expression in SAOS-2 and U2OS cells at 24 h, returning to baseline by 48 h. This suggests an early oxidative response; however, the lack of increases in SIRT1 and Nrf2 protein levels up to 48 h, as revealed by Western blot analysis, suggests an insufficient antioxidant defense, contributing to PD-induced cytotoxicity.

In our study, we also examined the effect of PD on human OS cell survival under stressful conditions mimicking the tumor microenvironment, specifically hypoxia, serum starvation, and chemotherapy. Hypoxia plays a key role in OS progression and resistance to therapy, contributing to increased cell proliferation, invasion, and angiogenesis [67,68]. Previous research has shown that resveratrol can counteract hypoxia-induced tumor aggression in SAOS-2 cells by downregulating HIF-1α [81]. Additionally, hypoxia promotes U2OS proliferation through the PDGF-BB/PDGFR-β axis [68], and contributes to DOX resistance in U2OS and MG-63 OS cells via AMP-activated protein kinase (AMPK) signaling [82]. Interestingly, our findings reveal for the first time that PD significantly inhibits OS cell growth under hypoxic conditions, particularly at 200 µM in SAOS-2 cells, and reduces U2OS growth at all tested doses. PD-treated cells also show increased sensitivity to serum starvation, suggesting that PD can enhance OS cell vulnerability even under challenging conditions where these cells typically become more resistant to treatments [66,69]. This led us to speculate that PD’s pro-oxidant effects may impair the adaptive mechanisms of SAOS-2 and U2OS cells by increasing ROS production or impairing ROS scavenging.

In this framework, we further evaluated the efficacy of PD in combination with the chemotherapeutic agents DOX and CIS, currently used in OS management [3]. These drugs are known to induce acute metabolic changes that may generate secondary harmful effects, such as endogenous ROS production and ferroptosis [57,62]. Studies suggest that dietary modulation of PD can increase the sensitivity to drugs like lapatinib, CIS, and DOX in various cancer types [71,83]. Additionally, PD can enhance the antitumor effects of CIS in NSCLC by inducing ROS-mediated ER stress [31]. In OS, PD enhances DOX-induced apoptosis in DOX-resistant OS, in vitro and in a MG-63/DOX xenograft model, by inhibiting Akt signaling through the oncogene taurine-upregulated gene 1 (TUG1) [26], and it improves paclitaxel sensitivity in U2OS and MG-63 cells [27]. Our preliminary data show that combined treatment of PD and DOX or CIS at their respective IC50 doses significantly increases cell sensitivity to these drugs. Notably, this is the first evidence that PD promotes cisplatin sensitivity in OS cells. It is reasonable to infer that the increased production of ROS may play a role in mediating these effects. As expected, cell growth inhibition is reduced in the presence of the ROS scavenger NAC, which is consistent with the effects observed with the individual treatments. This evidence underlines the potential of targeting ROS pathways in conjunction with PD to optimize therapeutic strategies for OS, warranting further investigation into their mechanisms and applications.

Overall, these results further support the potential of PD as a promising therapeutic agent for OS, demonstrating that PD-induced alterations in oxidative balance may contribute to its cytotoxic effects. However, there are some limitations to our study. The protective effects of polyphenols against cancer development remain a subject of ongoing debate, due to discrepancies between in vitro and clinical findings [12]. A more in-depth mechanistic exploration of PD’s pro-oxidant activity, particularly its role in inducing ferroptosis, along with studies assessing the long-term impact of PD treatment on OS cells, would provide a more thorough understanding of its therapeutic potential and durability. Additionally, the use of immortalized hFOB cells may have limitations, as their properties and responses to PD may differ from those of normal osteoblasts. To address this, further experiments on primary human osteoblasts are necessary to rule out potential non-specific toxicity at the high PD doses used in this study.

## 5. Conclusions

Overall, the results presented herein reveal for the first time that PD exerts its anti-cancer effects on SAOS-2 and U2OS cells at least partly by elevating ROS levels, depleting GSH stores, and increasing total iron levels, pushing the cells towards a state of oxidative stress. This redox imbalance increases the vulnerability of these cells to hostile conditions that cancer cells usually adapt to, such as hypoxia, nutrient deprivation, and exposure to the chemotherapeutic agents DOX and CIS. Although further research is needed to elucidate the underlying mechanisms that likely involve ferroptosis, these findings support the use of PD as an adjuvant therapy in the management of this aggressive bone tumor. It is noteworthy that PD is not toxic in animals up to a dose of 200 mg/kg and phase II clinical trials showed that it is also well tolerated in humans (40 mg twice a day for 90 days) [84,85]. However, as the therapeutic effects of PD on human subjects with bone and joint disorders have not yet been investigated [84], further studies, including those on potential clinical adverse events, are warranted.

## Figures and Tables

**Figure 1 cimb-47-00021-f001:**
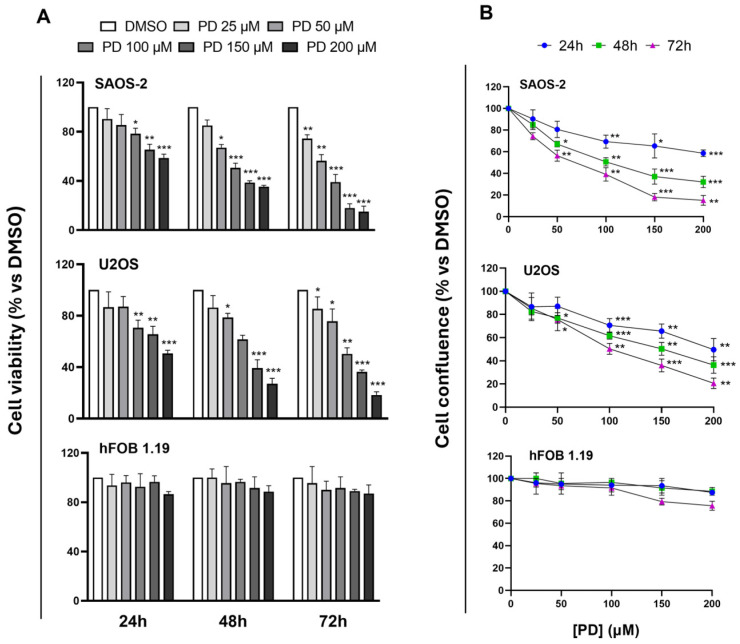
Effects of PD on growth and proliferation of SAOS-2, U2OS, and hFOB 1.19 cells. (**A**) The viability of SAOS-2, U2OS, and hFOB 1.19 cell lines was assessed by the MTT assay after treatment with PD (25–200 µM) or DMSO vehicle for the indicated time points. The percentage of cell viability of PD-treated cultures was calculated by normalizing their O.D. value to that of DMSO control cultures. (**B**) The proliferation of SAOS-2, U2OS, and hFOB 1.19 cell lines on type-I-collagen-pre-coated wells was evaluated by measuring live cell confluence after exposure to PD (25–200 µM) for 24, 48, and 72 h, using the confluence function of the Spark microplate reader. The percentage of cell confluence was calculated by normalizing values recorded for PD-treated cells to those of DMSO control cultures. All results are given as the mean ± SD of three independent experiments performed in triplicate. Statistical analysis was performed using one-way ANOVA followed by Tukey’s post hoc test, with significance levels indicated as * *p* < 0.05, ** *p* < 0.01, *** *p* < 0.001 compared with vehicle group.

**Figure 2 cimb-47-00021-f002:**
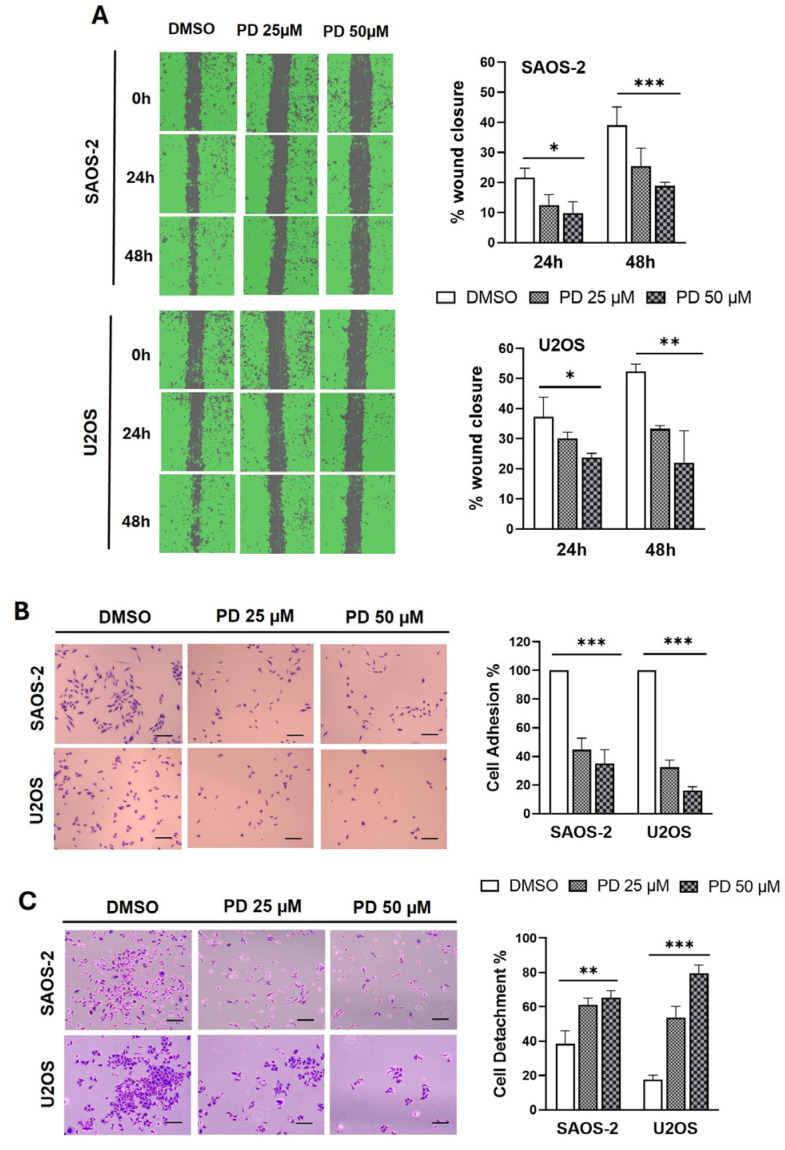
Effects of PD treatment on SAOS-2 and U2OS cell migration and adhesion/detachment capacity. (**A**) Comparison of cell migration between PD-treated cells (25 and 50 µM) and vehicle controls (DMSO ≤ 0.05%) using a scratch test, at the indicated time points. Quantitative analysis was performed by calculating the percentage change in the wound space after scratching (see Section 2 for further details). Left panels: Representative images of migrated SAOS-2 and U2OS cells recorded using the Tecan Spark instrument. Right panels: Histograms showing the percentage of wound closure. (**B**) Cell adhesion assays after 24 h incubation on pre-coated plates with PD (25 and 50 μM) or vehicle control. The attachment data were quantified as the percentage of attached cells in PD-treated samples compared to that in DMSO-treated samples. Left panel: Representative microscopical images of cell adhesion assays (scale bar: 100 μm), taken by an inverted light microscope (OLIMPUS EP 50). Right panels: The attachment data are quantified as the percentage of attached cells in PD-treated samples compared to that in DMSO-treated samples. (**C**) Cell detachment assays after 24 h incubation on pre-coated plates with PD (25 and 50 μM) or DMSO as control. Left panel: Representative microscopical pictures (scale bar: 100 μm). Right panels: Cell detachment data expressed as the percentage of attached cells after shaking vs. the initial cell number. All the results are given as the mean ± SD (n = 3). Statistical significance between treated and control groups was determined using one-way ANOVA followed by Tukey’s post hoc test with significance levels indicated as * *p* < 0.05, ** *p* < 0.01, *** *p* < 0.001.

**Figure 3 cimb-47-00021-f003:**
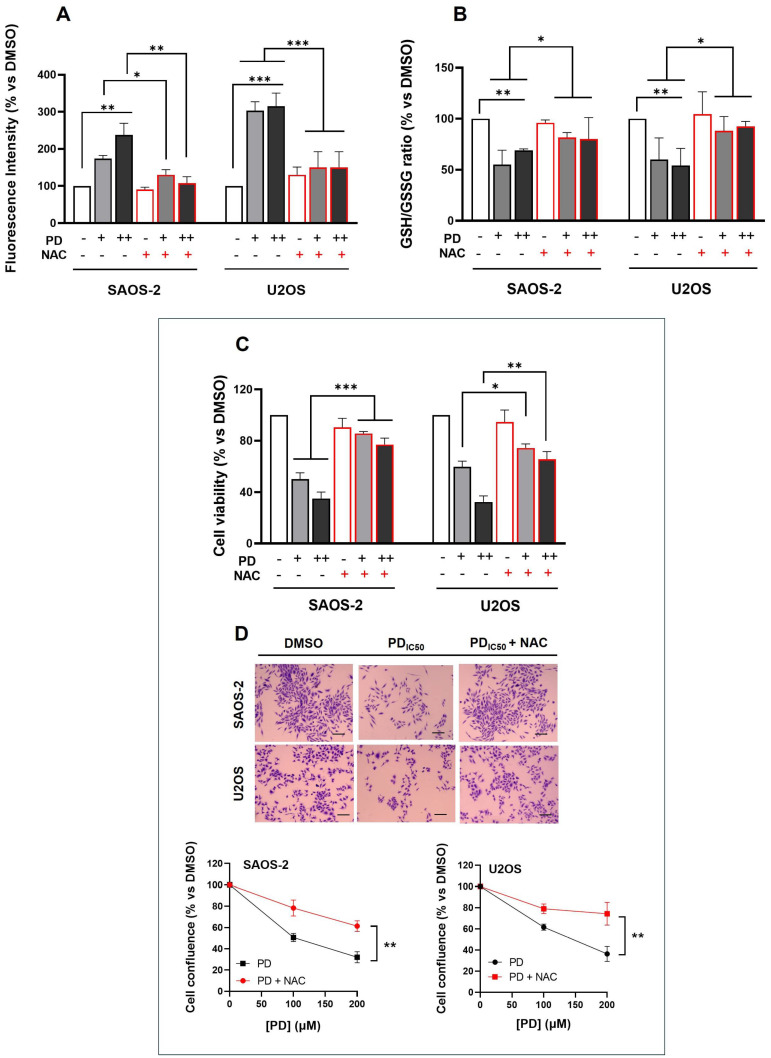
PD-induced changes in ROS and GSH levels in OS cells and their effect on cell viability and proliferation. SAOS-2 and U2OS cells were treated with PD 100 μM (+) and 200 μM (++) for 24 h (**A**,**B**) or 48 h (**C**,**D**). (**A**) DCF fluorescence assays showing the intracellular ROS production in SAOS-2 and U2OS cells, with or without pre-incubation with NAC (10 mM, 1 h prior to PD treatment). ROS are expressed as percentage of DCFDA fluorescence intensity compared to DMSO-treated cells arbitrarily set at 100%. (**B**) Cellular GSH/GSSG ratio measured by fluorometric microplate format. A quantity of 1 mM NAC was added or not to the growth medium for 1 h prior to PD treatment. Ratios are reported as percentage change compared to vehicle control. (**C**) Effects of ROS inhibition on cell viability assessed by the MTT assay after 48 h of treatment. The absorbance values at 570 nm were converted into percentage absorbance with respect to DMSO control. (**D**) Proliferation of SAOS-2 and U2OS cells on type-I-collagen-pre-coated wells, with or without pre-incubation with 10 mM NAC. Top panel: Representative images of SAOS-2 and U2OS cells stained with crystal violet after 48 h of treatment with PDIC50 (120 µM and 160 µM for SAOS-2 and U2OS, respectively) or DMSO; scale bar: 100 µm. Bottom panel: The percentage of cell live confluence of PD-treated cells at 48 h was measured using the confluence function of the Spark microplate reader and is reported as percentage vs. DMSO control cultures. All data are presented as the mean ± SD of three independent experiments with triplicate sets in each assay. Statistical significance between treatment groups and controls was determined using one-way ANOVA followed by Tukey’s post hoc test (** *p* < 0.01, *** *p* < 0.001). Comparisons between PD-treated samples (100 or 200 µM) and their respective NAC-treated samples were analyzed using one-way ANOVA followed by Tukey’s post hoc test (* *p* < 0.05, ** *p* < 0.01 and *** *p* < 0.001).

**Figure 4 cimb-47-00021-f004:**
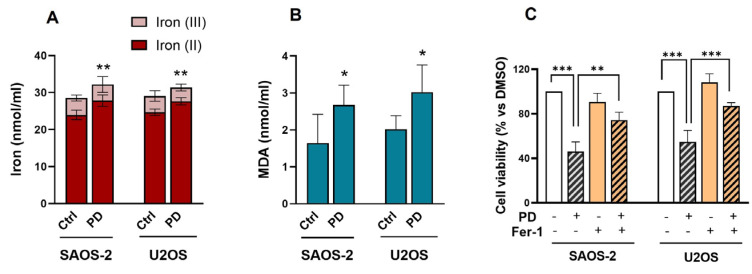
Impact of PD exposure on intracellular iron levels and MDA accumulation in OS cells. Total iron and oxidized Fe(II) and reduced Fe(III) in the cytosol of SAOS-2 and U2OS cells (**A**) and MDA levels (**B**) after 24 h treatment with PDIC50 values (120 µM and 160 µM for SAOS-2 and U2OS, respectively) or DMSO as control. Statistical significance was assessed using umpired *t*-tests (* *p* < 0.05 vs. DMSO). (**C**) Effects of Fer-1 on cell viability of PD-treated cells, assessed by the MTT assay. Cells were incubated with PD at their respective IC50 values, or vehicle control, in the presence or absence of Fer-1 (1 µM) for 48 h. One-way ANOVA followed by Tukey’s post hoc test was adopted to calculate the significant difference (** *p* < 0.01, *** *p* < 0.001). All data are given as the mean ± SD of two independent experiments performed in triplicate.

**Figure 5 cimb-47-00021-f005:**
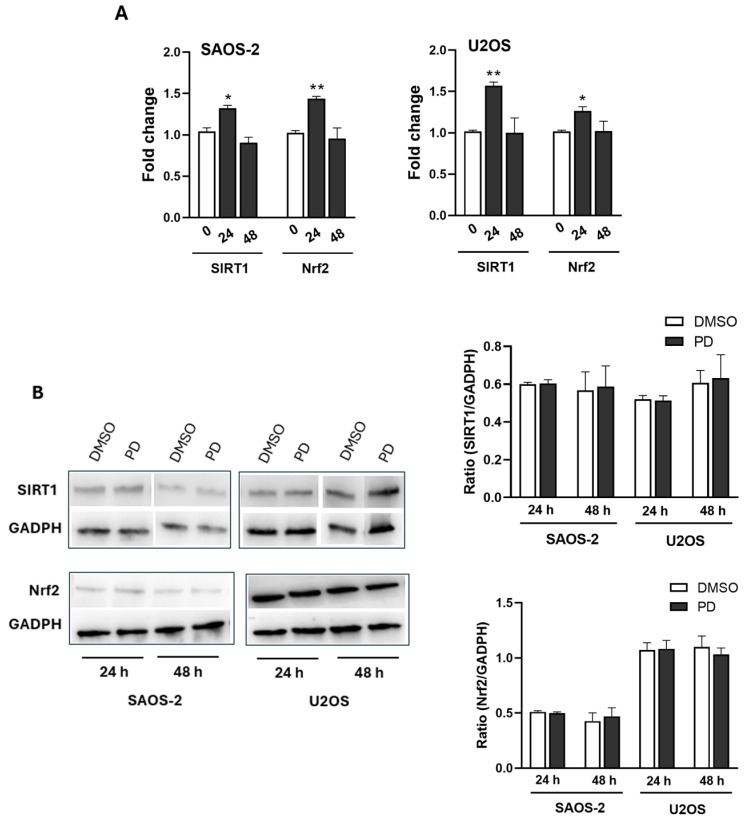
PD-induced effects on oxidative stress-related gene expression and protein levels. (**A**) The mRNA levels of SIRT-1 and Nrf-2 in SAOS-2 and U2OS cells were measured by real-time PCR after 24 and 48 h of treatment with PDIC50 values (120 µM and 160 µM for SAOS-2 and U2OS, respectively) or DMSO. The results are shown as means ± SD from two independent experiments with at least three technical replicates per condition. Significance between PD-treated and DMSO-treated samples was determined using one-way ANOVA, followed by Tukey’s post hoc test, with significance levels indicated as * *p* < 0.05, ** *p* < 0.01. (**B**) Western blot analysis. Left Panel: Blots probed with specific antibodies for SIRT1 (75 kDa), Nrf2 (68 kDa), and GAPDH (37 kDa). GAPDH was used as the loading control. Right Panel: The ratios were calculated following densitometric analysis of the bands from two independent experiments and expressed as mean ± SD.

**Figure 6 cimb-47-00021-f006:**
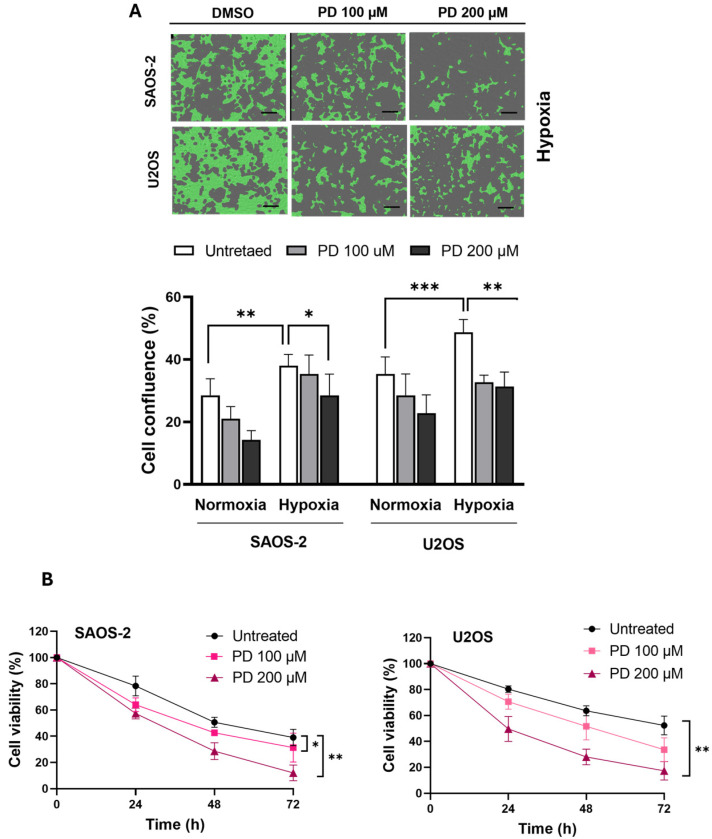
Effect of PD under conditions of hypoxia and serum starvation. (**A**) Effect of PD on the hypoxia-enhanced proliferation of SAOS-2 and U2OS cells. SAOS-2 and U2OS cells were incubated under normoxic or hypoxic conditions (37 °C, 5% CO_2_, 3% O_2_) for 24 h, either without or with the indicated concentrations of PD (3-h pre-treatment; see Section 2 for further details). Hypoxic cell culture conditions were maintained using the gas and humidity control function of the Tecan Spark instruments. Top Panel: Representative images of SAOS-2 and U2OS cells acquired under hypoxic condition, following 24 h of PD treatment (100 and 200 µM). Scale bar: 100 µm. Bottom Panel: The percentage of cell live confluence was measured using the same instruments. The results are shown as the mean ± SD of three independent experiments. Statistical significance was determined using one-way ANOVA followed by Tukey’s post hoc test (* *p* < 0.05, ** *p* < 0.01, *** *p* < 0.001). (**B**) Effects of PD treatment in OS cells grown under serum starvation conditions. SAOS-2 and U2OS were incubated in serum-reduced medium (0.5% FBS) with or without PD 100 and 200 µM, at 24, 48, and 72 h, prior to the MTT assay. The cell viability index was derived by dividing the O.D. at 570 nm at a given time over the O.D. recorded for the control cells (which were incubated without PD). Data shown are means ± SD of three independent experiments. * *p* < 0.05, ** *p* < 0.01, treated vs. untreated.

**Figure 7 cimb-47-00021-f007:**
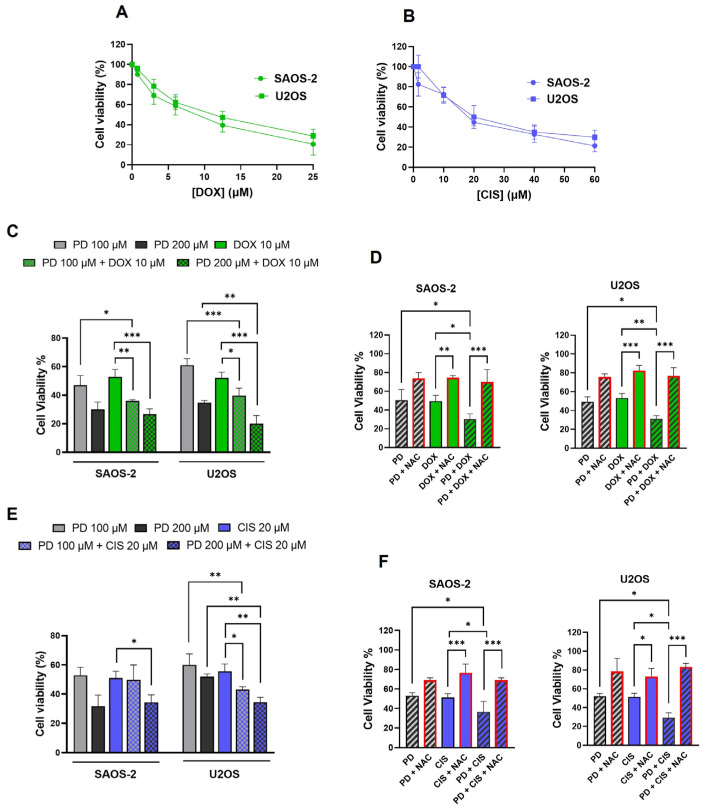
Combined effects of PD and chemotherapeutic agents, DOX and CIS, on OS cells growth. (**A**,**B**) SAOS-2 and U2OS were preliminarily treated with DOX (0.5–25 µM) or CIS (1.5–60 µM) for 48 h to determine the IC50 value for each drug. Cell viability is reported as absorbance results of MTT assay (Mean ± SD, n = 3). (**C**,**E**) Cytotoxicity effects of PD (100 or 200 µM) in combination with DOXIC50 (10 µM) or CISIC50 (20 µM) for 48 h, assessed by MTT assay. The data are reported as percentage values, based on optical density (O.D.) at 570 nm for cells treated with PD, DOX/CIS or DOX/CIS-PD co-treated, relative to the untreated control groups (set to 100%). Values represent mean ± SD of three independent experiments. Significance between different groups were determined using one-way ANOVA followed by Tukey’s post hoc test (* *p* < 0.05, ** *p* < 0.001, *** *p* < 0.001). (**D**,**F**) SAOS-2 and U2OS cells were pre-treated with 10 mM NAC for 1h before being exposed to PDIC50 (120 µM and 160 µM for SAOS-2 and U2OS, respectively), DOXIC50 or CISIC50 either individually or in combination with PDIC50/DOXIC50 or PDIC50/CISIC50, for 48 h. Data were analyzed using one-way ANOVA followed by Tukey’s multiple comparison test (* *p* < 0.05, ** *p* < 0.001, *** *p* < 0.001). Results are presented as the mean ± SD from three independent experiments. The untreated control group was set to 100%. Data for NAC-treated samples, previously reported in Figure 3, were omitted from the current graph.

## Data Availability

Data generated or analyzed during this study are provided in full within the published article.

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
