# Peer review of "Polydatin-Induced Shift of Redox Balance and Its Anti-Cancer Impact on Human Osteosarcoma Cells"

_cimb, 2024, doi:10.3390/cimb47010021_

Round 1

Reviewer 1 Report

Comments and Suggestions for Authors

Overall, this is a well-written manuscript, and the main goal is thoroughly explained. The authors put a lot of work and time into article preparation. The introduction is clear and presents information about the previous study findings that allows the reader to follow through the researchers' thinking. The methodology is well written and includes many details that are important for its replicability and/or reproducibility. The methods used in this study are appropriate, the controls are well selected, and the authors considered many factors that will be important in osteosarcoma cell therapy. The results are comprehensible and consistent with the observations presented by other research groups. This article makes a significant contribution to the research literature, especially considering the complexity of osteosarcoma mutagenesis and chemotherapy resistance pathways. Specific comments are presented in the following.

1.      In line 142 (5 mg/mL in PBS w/w Ca2+ and Mg2). It should probably be w/o

2.      According to the literature, the wound healing assay should be performed for a maximum of 24 hours. If you monitor migration for up to 48 hours, you should use mitomycin C to inhibit proliferation. In this study, the authors did not use mitomycin C. For that reason, I suggest adding the information that the wound healing assay was used to monitor the migration and proliferation of OS cells.

3.      I am wondering, did the authors use any cell detachment reagent (e.g. trypsin)? Because it was not mentioned in the methodology (2.6 Detachment assay description).

4.      Please change the marks in Figures 1A, B. There is no need to use # or x if you analyze statistical significance between control and treated cell. Other marks will be helpful if you compare, for example 25µM of PD vs 200µM of PD.

5.      In lines 528-531 the authors mentioned that: ‘Interestingly, Fer-1 treatment significantly prevented PD-induced cell death in both SAOS-2 and U2OS cells (Figure 4C), with a particularly strong effect in U2OS (p < 0.001), suggesting that ferroptosis may play a role in the cytotoxic effects of PD.’ I suggest changing this sentence, for example: Interestingly, Fer-1 treatment significantly prevented PD-induced cell death in both SAOS-2 and U2OS cells (Figure 4C), with a particularly strong effect in U2OS (p < 0.001), suggesting that PD may promote cytotoxic effects via ferroptosis induction’ or something similar.  

6.      Please edit figure 6B (x-axis Time [h], not in [hrs]). This mark is used more often.  

7.      The authors observed an increase in the expression of the SIRT1 and Nrf2 genes 24 h after treatment. However, regulation of protein level is delayed in time. Therefore, SIRT1 and Nrf2 protein levels should be analyzed 24- and 48 h after PD treatment. I recommend changing the sentence in lines 758 – 760 (This suggests an early oxidative response, but the lack of sustained increases in SIRT1 and Nrf2 protein levels points to an insufficient antioxidant defense, contributing to PD-induced cytotoxicity) because, in my opinion, it is a little misleading.

Reviewer 2 Report

Comments and Suggestions for Authors
  • This paper is well-organized and articulately written, addressing critical issues in cancer , but I do have some questions

  • Could the author provide a reference for the cell viability assays mentioned in section 2.2? Typically, DMSO is used to extract formazan. For instance, a volume of 100 μL/well of extraction buffer (5% SDS in N,N-Dimethylformamide) was added. It is crucial to cite references for such assays, including those used for proliferation and wound healing.

  • Is it advisable to use a 96-well format for the wound healing assay? Considering the potential space limitations in such wells, is there sufficient area for a scratch to fully recover? Additionally, the manuscript does not specify the number of cells seeded, which is vital for reproducibility.

  • The concentration of N-acetylcysteine (NAC) mentioned is 10 mM. This seems potentially high—could the author clarify if this might excessively suppress cell function? Moreover, there is a concern about possible fluorescence saturation in ROS detection. Would it be more informative to investigate ROS levels at different detection times to ensure accurate readings?

  • For the iron detection assays, does the medium itself contain iron? If so, it might be necessary to use an iron-free medium to avoid interference with the assay results. Additionally, considering the potential for ferrous ions to catalyze Fenton-like reactions, which can impact cell viability, it would be beneficial if the author could investigate a range of doses, both lower and higher than the IC50. This could provide a clearer understanding of the iron's impact on cell survival across different concentrations.
